Genome-wide analysis of the WRKY genes and their important roles during cold stress in white clover

Li Manman
Zhang Xueqi
Zhang Tianxiang
Bai Yan
http://orcid.org/0000-0002-1865-7751 Chen Chao
Guo Donglin
Guo Changhong kaku3008@126.com
http://orcid.org/0000-0003-4463-0218 Shu Yongjun syjun2003@126.com
College of Life Science and Techonology, Harbin Normal University , Harbin, Heilongjiang , China
Zhang Lin
Electronic publication date: 2023 Jul 11
Publication date: 2023
Volume: 11
Electronic Location ID: e15610
Received 2023 Mar 13; Accepted 2023 May 31
Copyright: © 2023 Li et al.
Copyright year: 2023
Copyright holder: Li et al.
License: This is an open access article distributed under the terms of the Creative Commons Attribution License, which permits unrestricted use, distribution, reproduction and adaptation in any medium and for any purpose provided that it is properly attributed. For attribution, the original author(s), title, publication source (PeerJ) and either DOI or URL of the article must be cited.
License URL: https://creativecommons.org/licenses/by/4.0/

Keywords: White clover, WRKY, Genetic regulation network, Cold stress

Funding: Natural Science Foundation of Heilongjiang Province LH2022C050 Harbin Normal University HSDSSCX2022-31 Natural and Science Foundation of China U21A20182 China Postdoctoral Science Foundation 2022M711431 This work was supported by the Natural Science Foundation of Heilongjiang Province (grant number LH2022C050), the Innovative Project for Postgraduate Students of Harbin Normal University (grant number HSDSSCX2022-31), the Natural and Science Foundation of China (grant number U21A20182), the China Postdoctoral Science Foundation (grant number 2022M711431). The funders had no role in study design, data collection and analysis, decision to publish, or preparation of the manuscript.

==============================
Background

White clover (Trifolium repens L) is a high-quality forage grass with a high protein content, but it is vulnerable to cold stress, which can negatively affect its growth and development. WRKY transcription factor is a family of plant transcription factors found mainly in higher plants and plays an important role in plant growth, development, and stress response. Although WRKY transcription factors have been studied extensively in other plants, it has been less studied in white clover.

Methods and Results

In the present research, we have performed a genome-wide analysis of the WRKY gene family of white clover, in total, there were 145 members of WRKY transcription factors identified in white clover. The characterization of the TrWRKY genes was detailed, including conserved motif analysis, phylogenetic analysis, and gene duplication analysis, which have provided a better understanding of the structure and evolution of the TrWRKY genes in white clover. Meanwhile, the genetic regulation network (GRN) containing TrWRKY genes was reconstructed, and Gene Ontology (GO) annotation analysis of these function genes showed they contributed to regulation of transcription process, response to wounding, and phosphorylay signal transduction system, all of which were important processes in response to abiotic stress. To determine the TrWRKY genes function under cold stress, the RNA-seq dataset was analyzed; most of TrWRKY genes were highly upregulated in response to cold stress, particularly in the early stages of cold stress. These results were validated by qRT-PCR experiment, implying they are involved in various gene regulation pathways in response to cold stress.

Conclusion

The results of this study provide insights that will be useful for further functional analyses of TrWRKY genes in response to biotic or abiotic stresses in white clover. These findings are likely to be useful for further research on the functions of TrWRKY genes and their role in response to cold stress, which is important to understand the molecular mechanism of cold tolerance in white clover and improve its cold tolerance.

Introduction

During the life cycle of plants on Earth, they often encounter various stresses, seriously hindering their growth and development (Chinnusamy, Schumaker & Zhu, 2004; Katagiri, 2004). Among these stresses, most of them are abiotic stresses, such as drought, salt, and cold, etc. In long-term evolution, plants have gradually developed numerous molecular regulatory mechanisms to confer various stresses. These regulatory mechanisms have employed many genes involved in complex regulatory processes, which adjust plants’ physiological and biochemical processes in adapting to adverse environments. Among these genes, transcription factors (TF) are richly distributed in plants, which play very important roles in response to stress (Singh, Foley & Onate-Sanchez, 2002; Yamaguchi-Shinozaki & Shinozaki, 2006). The TF genes can bind to DNA regions, which are named as cis-acting elements, and regulate downstream gene expression. According to kinds of DNA regions, TF genes are divided into many classes, such as AP2/ERF, bZIP, WRKY, MYB, bHLH, etc., (Singh, Foley & Onate-Sanchez, 2002).

The WRKY gene family is a transcription factor family that has been specifically and widely identified in plants, mainly in higher plants, but rarely in lower organisms (Eulgem et al., 2000). These WRKY TFs can bind to cis-acting element (named as W-box, (T)(T)TGAC(C/T)), and regulate expression of downstream target genes containing W-box in promotors. These genes are characterized with critical roles in response to biotic, abiotic, and hormonal signaling processes (Ülker & Somssich, 2004). The WRKY gene was first isolated from sweet potato and named Sweet Potato Factor 1 (SPF1) (Ishiguro & Nakamura, 1994), it has since been widely identified and characterized from a lot of plants, and has become one of the largest TF families in plants, such as 72 WRKY TFs in Arabidopsis thaliana and 102 WRKY members in rice (Oryza sativa) (Abdullah-Zawawi et al., 2021). With the development of genome sequencing projects, many WRKY TFs have been well identified in dozens of plants with completed genome sequencing, such as cucumber (Cucumis sativus L) (Chen et al., 2020b), poplar (Populus tremula), grape (Vitis vinifera) (Wang et al., 2014), alfalfa (Medicago sativa L) (Mao et al., 2020), tea tree (Camellia sinensis), etc., (Chen et al., 2015, 2016; Rinerson et al., 2015; Singh et al., 2019; Xie et al., 2018). The roles of WRKY TFs have been widely documented in regulating plant growth, development process, and especially in response to abiotic stresses (Li et al., 2020). Yoo et al. (2014) have found that multiple WRKY genes could modulate response to abiotic stress, while over-expression of OsWRKY genes would enhance its tolerance to abiotic stress, and many WRKY TFs have also been characterized with a high capability to improve abiotic stress tolerance in various plants (Ülker & Somssich, 2004).

White clover (Trifolium repens L) is a perennial legume plant which is widely distributed in temperate and cool-temperate regions due to its strong root development, prostrate growth, and rapid regeneration (Wu et al., 2021). It also has a high yield, good quality, and adaptability, making it an important economic crop and an ideal forage for reforestation and improvement of natural grassland in some regions, which is also widely used as a landscape plant in gardens. However, during its growth process, white clover inevitably encounters various abiotic stresses, such as high salt, drought, and cold stress, especially in hard-winter of high-latitude regions, which causes extremely abnormal death of white clover and severely affects their production and promotion. Therefore, improving the cold tolerance of white clover has become an incisive problem in white clover production, while its genetic knowledge is still poor (Bao et al., 2020; Inostroza et al., 2018). Until 2019, the genome sequences of white clover were released, and researchers have been able to investigate gene function at the genome-level, which greatly promoted genetic improvement works in white clover (Griffiths et al., 2019). For example, Ma et al. (2022) have identified 37 SPL transcription factors from the white clover genome and characterized their key functions in inflorescence development.

Based on these results, our study has used bioinformatics methods to identify WRKY TFs from white clover at the genome-wide level, and systematically characterized their structure constitutions, cis-acting elements analysis, chromosome distributions, and genetic regulation network by integrating various datasets. Furthermore, RNA-seq was adopted to investigate WRKY TFs in response to cold stress, and qRT-PCR also confirmed their expression profiles. These findings would provide valuable insights into the exploration of white clover WRKY functions in response to cold stress.

Materials and Methods

Identification and classification of the TrWRKY gene family in white clover

The white clover genome resource information was released from the previous study, and all files were provided by Stig Uggerhøj Andersen from Aarhus University (Griffiths et al., 2019). DNA, CDS, and protein sequences were retrieved from the white clover genome. The sequences of Arabidopsis WRKY family proteins were collected from the TAIR database and used as BLAST (version 2.9.0+) query sequences to search the genome of white clover (Altschul et al., 1997), with an evaluation setting of 1E-05 and the coverages were set as 80%. The HMM file (version 3.3) (PF03106) was downloaded from the Pfam database (Punta et al., 2011), and the HMMER (evalue: 0.01) was used to identify and confirm WRKY DNA-binding domain, which was characterized as candidate WRKY proteins (Finn, Clements & Eddy, 2011). Annotated information for all candidate WRKY genes was retrieved from the white clover genome, including genome position, protein length, intron numbers, and these WRKY genes were classified into groups based on similar WRKY genes in Arabidopsis.

Phylogenetic analysis of the TrWRKY genes in white clover

Multiple sequence alignment analysis of Arabidopsis thaliana and white clover WRKY proteins using MUSCLE (version 5.1.0) with default parameters (Edgar, 2004). The phylogenetic tree of the WRKY gene family was constructed using the MEGA (version 11; Tamura, Stecher & Kumar, 2021) method with the following parameters: (1) Neighbor-joining (NJ); (2) Poisson correction; (3) genetic distance; (4) pair-wise deletion; (5) bootstrap: 1,000 replications. The TrWRKY genes were classified into different groups and subgroups based on the phylogenetic tree of AtWRKY and TrWRKY sequences.

Motif composition distribution analysis of TrWRKY genes in white clover

The conserved motifs of white clover WRKY protein sequences were identified using MEME (Multiple EM for motif Elicitation, Version 4.8.1) with the following parameters: (1) minimum and maximum width: 10 and 50, respectively; (2) the maximum number of motifs, 10; (3) number of appearances of a single pattern distributed in the sequences with model: 0 or 1 per sequence (-modzoops) (Bailey et al., 2009). All results were displayed with TBtools (version 1.098; Chen et al., 2020a).

Cis-acting elements analysis in the promoters of members of the TrWRKYs gene family

The 1,000 bp genomic sequence upstream of the transcription start site of the WRKYs gene family members was obtained from the white clover genome and the cis-acting elements in the promoter region of TrWRKYs gene family members were predicted by the PlantCARE online tool (https://bioinformatics.psb.ugent.be/webtools/plantcare/html/).

Chromosomal location, gene duplication and Ka/Ks analysis of WRKY genes in white clover

All white clover proteins were compared to each other using the software BLASTP (version 2.9.0+), and the gene duplications were identified and characterized based on BLAST results by software MCSanX (version Python) with the default parameter (Wang et al., 2012). Based on the positional information of WRKY genes in the white clover genome and the duplication between genes, the software CIRCOS (version 0.69-8) was used to display the distribution of the WRKY gene family in the white clover genome (Krzywinski et al., 2009). Based on the phylogenetic tree and gene duplications results, a molecular evolutionary analysis of the TrWRKY genes was performed by calculating the nonsynonymous (Ka) to synonymous (Ks) substitution ratio of the duplicated gene pairs in S. tuberosum using the KaKs_Calculator in TBtools (Chen et al., 2020a).

Gene regulation network analysis of white clover WRKY gene family

The information on the Arabidopsis gene regulatory network (GRN) was extracted from the AraNet database (V2) (Lee et al., 2014), including 22,894 Arabidopsis genes and 895,000 interactions (links). All white clover proteins were subjected to BLAST searched with Arabidopsis proteins, with an e-value cut-off 1e-05, and the highest scoring hits were confirmed as homologous genes of the white clover gene. Meanwhile, all Arabidopsis proteins were also BLAST searched against white clover proteins with the same set, and hits with the highest scores were identified as homologous genes for Arabidopsis genes; the two genes were identified as homologous pairs with two BLAST results. Then, the GRN of white clover was constructed using the GRN of Arabidopsis based on homologous pairs. The sub-networks that contained the TrWRKY gene of white clover were searched and evaluated; the results were visualized using Cytoscape software (version 3.9.1), as well as the genes were annotated using GO information based on white clover genome annotation information (Shannon et al., 2003). Then, GO enrichment analysis of the sub-networks was performed using topGO software (version 2.50.0; Alexa & Rahnenfuhrer, 2019), the threshold level was set to 0.05 to show the most significant terms, and terms with high enrichment were assigned as GRN functions as described in the software protocol.

Expression analysis of white clover TrWRKY genes in response to cold stress

The RNA-seq data have been reported, the data included eight time points in response to cold stress; our previous work has described in detail, which could be assessed with accession numbers: PRJNA781064 (Zhang et al., 2022). All of the RNA-seq reads were mapped to the transcript sequences of white clover genomes using the Salmon software (version 0.12.0), and the expression level of each gene (FPKM value) was estimated using the subroutine quant of software Salmon (Patro et al., 2017). These expressional data were transformed using the “log2” function and they were centered using the “scale” function of the R program (version 4.2.1; R Core Team, 2022); then, all expression data were clustered and plotted using the “heatmap.2” function of the ggplots package (version 3.1.3).

Plant growth and qRT-PCR analysis

Seeds of white clover cv. Haifa were purchased from Barenbrug China Ltd. Com. (Beijing, China). All seeds were germinated and transferred to a mixture of perlite and sand each with a volume of 3:1, as our previously described (Zhang et al., 2022). In briefly, all seeds were growing in the pots, about 10–15 plants per pot. The growth temperature was 24 °C in light and 18 °C in darkness per day and irrigated with half-strength Hoagland solution once every 2 days. After 4 weeks, they were randomly divided into four groups for cold stress treatment. We collected at 0 min (control), 30 min, 1 h, and 3 h (4 time points in total) at a setting of 4 °C. For each group, three samples were randomly chosen of five seedlings were pooled to form a biological replicate. All samples were frozen in liquid nitrogen and stored at −80 °C. Total RNA was extracted from white clover seedlings at different time points of cold stress at 4 °C using the Total Plant RNA Extraction Kit (Tiangen, Beijing, China), divided into four time points (0 min (control), 30 min, 1 h, 3 h), and used Prime Script RT kit (Toyobo, Shanghai, China) was reverse transcribed into cDNA as a template for quantitative reverse transcription PCR. Primers were designed using Primer3 based on the nucleotide sequences of WRKY family genes (Table S1) (Untergasser et al., 2012). qRT-PCR was performed using a Light Cycler® 96 system (Roche, Rotkreuz, Switzerland) and SYBR Premix Ex TaqTMII (Toyobo, Shanghai, China). Three replicates of each experiment were performed. The PCR conditions were set as follows. 95 °C for 2 min, 40 cycles, 95 °C for 30 s, 55 °C for 30 s, and 72 °C for 1 min. Fold change values were calculated using expression abundance, which is based on the 2−ΔΔCT method.

Results and analysis

Identification of WRKY genes in white clover

A total of 145 TrWRKY genes were successfully identified from the white clover genome by multiple sequence alignment based on the amino acid sequence of the white clover WRKY gene family. These genes were proven to contain WRKY domains according to domain analysis and they were named TrWRKY1 to TrWRKY145 according to their chromosome locus and structure. All genomic information of these TrWRKY genes, including names, gene locus, chromosomal locations, group, introns and protein length (aa) were retrieved and summarized in Table 1. For these 145 TrWRKY genes, the largest protein was TrWRKY145, comprising 1,063 amino acids (aa), while the smallest one was TrWRKY65 (137aa). There are 16 chromosomes in the white clover genome; the TrWRKY genes were not evenly distributed across all chromosomes. The intron distribution was 1—11, of which the TrWRKY121 had the most introns (11 introns), while fifteen members contained only one intron; the results showed TrWRKY genes highly diverged.

Table 1 Summary of TrWRKY genes identified in the white clover.

Name	Locus	Chromosomal locations	Group	Intron	Length (aa)	
TrWRKY001	chr1.jg4946	Tr1O:35253397–35255542	I	4	155	
TrWRKY002	chr5.jg5422	Tr5O:36224162–36225107	I	1	195	
TrWRKY003	chr1.jg4948	Tr1O:35276740–35279067	I	2	206	
TrWRKY004	chr5.jg4677	Tr5O:31183372–31185712	I	3	290	
TrWRKY005	chr7.jg5842	Tr7O:36085774–36087462	I	2	340	
TrWRKY006	chr12.jg3166	Tr4P:21046044–21048271	I	3	378	
TrWRKY007	chr2.jg5856	Tr2O:39405643–39408558	I	3	379	
TrWRKY008	chr3.jg9578	Tr3O:62913893–62916736	I	3	425	
TrWRKY009	chr15.jg2764	Tr7P:18230475–18233187	I	4	436	
TrWRKY010	chr9.jg5404	Tr1P:36641010–36643525	I	4	441	
TrWRKY011	chr4.jg377	Tr4O:2750219–2753428	I	4	462	
TrWRKY012	chr4.jg517	Tr4O:3682810–3686140	I	4	462	
TrWRKY013	chr13.jg5881	Tr5P:38784060–38785660	I	2	470	
TrWRKY014	chr9.jg4026	Tr1P:27139671–27144775	I	4	487	
TrWRKY015	chr16.jg6482	Tr8P:46034663–46039091	I	4	493	
TrWRKY016	chr3.jg1725	Tr3O:11498359–11502499	I	4	507	
TrWRKY017	chr3.jg810	Tr3O:5276114–5280219	I	4	511	
TrWRKY018	chr7.jg4527	Tr7O:28275075–28279600	I	3	517	
TrWRKY019	chr16.jg6484	Tr8P:46056816–46061810	I	5	522	
TrWRKY020	chr13.jg5948	Tr5P:39165179–39168802	I	5	536	
TrWRKY021	chr7.jg2559	Tr7O:16132786–16135642	I	4	542	
TrWRKY022	chr15.jg1220	Tr7P:8126860–8129794	I	4	546	
TrWRKY023	chr11.jg2218	Tr3P:14403172–14409407	I	5	550	
TrWRKY024	chr5.jg6632	Tr5O:44854411–44858046	I	5	550	
TrWRKY025	chr13.jg5557	Tr5P:36394815–36398400	I	4	566	
TrWRKY026	chr3.jg3718	Tr3O:24080424–24083938	I	4	568	
TrWRKY027	chr3.jg9591	Tr3O:62992430–62995677	I	4	576	
TrWRKY028	chr13.jg5883	Tr5P:38801923–38805881	I	4	611	
TrWRKY029	chr7.jg4480	Tr7O:27919168–27923981	I	7	625	
TrWRKY030	chr7.jg4830	Tr7O:30168002–30171661	I	4	659	
TrWRKY031	chr3.jg54	Tr3O:364289–368532	I	4	661	
TrWRKY032	chr13.jg5616	Tr5P:36858900–36863941	I	4	667	
TrWRKY033	chr9.jg1959	Tr1P:13473819–13478644	I	4	668	
TrWRKY034	chr1.jg2939	Tr1O:21500495–21505105	I	4	669	
TrWRKY035	chr14.jg4072	Tr6P:26714269–26720795	I	6	897	
TrWRKY036	chr1.jg1973	Tr1O:14366719–14368228	IIa	3	249	
TrWRKY037	chr9.jg1140	Tr1P:7918363–7919974	IIa	3	249	
TrWRKY038	chr1.jg1972	Tr1O:14356694–14358566	IIa	3	284	
TrWRKY039	chr9.jg1139	Tr1P:7914343–7916022	IIa	3	285	
TrWRKY040	chr1.jg2956	Tr1O:21601386–21603577	IIa	3	288	
TrWRKY041	chr10.jg540	Tr2P:3685109–3687534	IIa	3	288	
TrWRKY042	chr12.jg1297	Tr4P:8797774–8800048	IIa	4	314	
TrWRKY043	chr4.jg283	Tr4O:2101646–2103890	IIa	4	314	
TrWRKY044	chr14.jg4718	Tr6P:31206787–31208142	IIa	3	321	
TrWRKY045	chr6.jg5830	Tr6O:39546655–39548025	IIa	3	322	
TrWRKY046	chr3.jg2104	Tr3O:14203754–14206210	IIb	4	367	
TrWRKY047	chr6.jg3356	Tr6O:22856414–22858929	IIb	3	388	
TrWRKY048	chr3.jg2106	Tr3O:14209147–14217688	IIb	7	407	
TrWRKY049	chr11.jg4323	Tr3P:28623299–28628203	IIb	5	420	
TrWRKY050	chr13.jg6040	Tr5P:39734181–39741546	IIb	5	445	
TrWRKY051	chr5.jg5942	Tr5O:39784978–39789205	IIb	5	450	
TrWRKY052	chr4.jg9125	Tr4O:64569075–64575507	IIb	6	473	
TrWRKY053	chr3.jg4106	Tr3O:26548251–26551282	IIb	3	494	
TrWRKY054	chr8.jg8224	Tr8O:57583206–57585522	IIb	5	496	
TrWRKY055	chr8.jg8226	Tr8O:57595609–57597977	IIb	5	496	
TrWRKY056	chr11.jg6061	Tr3P:40558111–40560701	IIb	4	497	
TrWRKY057	chr9.jg3992	Tr1P:26841184–26844303	IIb	3	501	
TrWRKY058	chr14.jg619	Tr6P:4268713–4272199	IIb	3	521	
TrWRKY059	chr4.jg1879	Tr4O:13724492–13727623	IIb	4	556	
TrWRKY060	chr8.jg7568	Tr8O:52902921–52906615	IIb	4	556	
TrWRKY061	chr10.jg1880	Tr2P:13055576–13058377	IIb	5	564	
TrWRKY062	chr10.jg1240	Tr2P:8565304–8567908	IIb	5	565	
TrWRKY063	chr6.jg3259	Tr6O:22013991–22017015	IIb	3	573	
TrWRKY064	chr4.jg5478	Tr4O:39794266–39798649	IIb	4	611	
TrWRKY065	chr14.jg1701	Tr6P:11603027–11604397	IIc	2	137	
TrWRKY066	chr9.jg1339	Tr1P:9315424–9316217	IIc	2	152	
TrWRKY067	chr16.jg6071	Tr8P:43479876–43480502	IIc	1	172	
TrWRKY068	chr9.jg4957	Tr1P:33609237–33616647	IIc	1	188	
TrWRKY069	chr3.jg7827	Tr3O:50998463–50999731	IIc	1	191	
TrWRKY070	chr12.jg4938	Tr4P:32185999–32188206	IIc	1	215	
TrWRKY071	chr3.jg7996	Tr3O:52089540–52090962	IIc	1	229	
TrWRKY072	chr8.jg6405	Tr8O:44781392–44782814	IIc	1	229	
TrWRKY073	chr7.jg896	Tr7O:5551407–5554787	IIc	2	241	
TrWRKY074	chr16.jg4775	Tr8P:33960705–33961961	IIc	3	242	
TrWRKY075	chr11.jg7038	Tr3P:46907394–46909934	IIc	2	246	
TrWRKY076	chr8.jg6410	Tr8O:44809752–44812152	IIc	2	250	
TrWRKY077	chr14.jg1115	Tr6P:7734201–7741598	IIc	4	266	
TrWRKY078	chr2.jg886	Tr2O:6217545–6220827	IIc	2	273	
TrWRKY079	chr7.jg7388	Tr7O:46299748–46301293	IIc	2	289	
TrWRKY080	chr3.jg7829	Tr3O:51008820–51011087	IIc	2	313	
TrWRKY081	chr4.jg7787	Tr4O:55224476–55225976	IIc	2	318	
TrWRKY082	chr1.jg12558	Tr1O:86758945–86760457	IIc	2	323	
TrWRKY083	chr12.jg7469	Tr4P:49027733–49032235	IIc	6	343	
TrWRKY084	chr1.jg10510	Tr1O:72595887–72597616	IIc	2	358	
TrWRKY085	chr10.jg2272	Tr2P:15800686–15803310	IIc	2	368	
TrWRKY086	chr16.jg5753	Tr8P:41361956–41363810	IIc	2	404	
TrWRKY087	chr8.jg8112	Tr8O:56771672–56773457	IIc	2	406	
TrWRKY088	chr14.jg1447	Tr6P:9967786–9968685	IId	1	149	
TrWRKY089	chr16.jg297	Tr8P:1981047–1981864	IId	1	153	
TrWRKY090	chr5.jg6489	Tr5O:43735385–43739232	IId	1	165	
TrWRKY091	chr7.jg1916	Tr7O:11673947–11677464	IId	1	165	
TrWRKY092	chr3.jg10600	Tr3O:70128861–70129969	IId	2	223	
TrWRKY093	chr3.jg376	Tr3O:2419836–2420993	IId	2	245	
TrWRKY094	chr5.jg633	Tr5O:4407752–4409399	IId	2	250	
TrWRKY095	chr3.jg418	Tr3O:2696699–2704267	IId	6	258	
TrWRKY096	chr11.jg9390	Tr3P:63448611–63450828	IId	3	260	
TrWRKY097	chr8.jg8664	Tr8O:60972016–60973116	IId	2	290	
TrWRKY098	chr12.jg3044	Tr4P:20303684–20304867	IId	2	314	
TrWRKY099	chr4.jg8047	Tr4O:57107571–57109224	IId	2	327	
TrWRKY100	chr9.jg1028	Tr1P:7138533–7141172	IId	2	345	
TrWRKY101	chr9.jg954	Tr1P:6662343–6664904	IId	2	349	
TrWRKY102	chr9.jg5416	Tr1P:36708761–36713161	IId	3	350	
TrWRKY103	chr8.jg7125	Tr8O:49744559–49746242	IIe	1	154	
TrWRKY104	chr10.jg758	Tr2P:5125509–5126589	IIe	2	215	
TrWRKY105	chr8.jg6926	Tr8O:48516893–48517971	IIe	2	215	
TrWRKY106	chr4.jg10192	Tr4O:72095432–72096468	IIe	2	237	
TrWRKY107	chr7.jg7373	Tr7O:46208011–46209900	IIe	2	257	
TrWRKY108	chr15.jg4042	Tr7P:26578475–26580507	IIe	2	259	
TrWRKY109	chr6.jg179	Tr6O:1356330–1358362	IIe	2	259	
TrWRKY110	chr13.jg4575	Tr5P:30183713–30187166	IIe	2	262	
TrWRKY111	chr4.jg3667	Tr4O:26832093–26835148	IIe	2	262	
TrWRKY112	chr9.jg5891	Tr1P:39840694–39844509	IIe	5	277	
TrWRKY113	chr4.jg8395	Tr4O:59509470–59511001	IIe	1	296	
TrWRKY114	chr1.jg12229	Tr1O:84484169–84486864	IIe	2	307	
TrWRKY115	chr9.jg7902	Tr1P:53219706–53221002	IIe	2	317	
TrWRKY116	chr16.jg1080	Tr8P:7667356–7669114	IIe	2	324	
TrWRKY117	chr16.jg3334	Tr8P:23935985–23937743	IIe	2	324	
TrWRKY118	chr8.jg542	Tr8O:4059377–4061115	IIe	2	329	
TrWRKY119	chr1.jg7088	Tr1O:49874567–49875820	IIe	2	337	
TrWRKY120	chr7.jg3290	Tr7O:20634234–20635490	IIe	2	337	
TrWRKY121	chr1.jg8379	Tr1O:58443647–58446096	IIe	2	370	
TrWRKY122	chr6.jg3054	Tr6O:20633120–20636355	IIe	2	483	
TrWRKY123	chr10.jg1494	Tr2P:10329830–10331679	IIe	2	517	
TrWRKY124	chr12.jg6707	Tr4P:43890308–43904513	IIe	11	919	
TrWRKY125	chr14.jg2719	Tr6P:18441055–18442214	III	1	182	
TrWRKY126	chr3.jg6078	Tr3O:40310959–40312118	III	1	182	
TrWRKY127	chr16.jg69	Tr8P:454817–456279	III	2	242	
TrWRKY128	chr3.jg8340	Tr3O:54563576–54565215	III	2	263	
TrWRKY129	chr1.jg1680	Tr1O:12254543–12257169	III	2	302	
TrWRKY130	chr9.jg3115	Tr1P:20932379–20934360	III	2	305	
TrWRKY131	chr9.jg6983	Tr1P:47150282–47152529	III	3	307	
TrWRKY132	chr16.jg1458	Tr8P:10549490–10551175	III	2	309	
TrWRKY133	chr2.jg4287	Tr2O:28961889–28963324	III	2	321	
TrWRKY134	chr4.jg11883	Tr4O:83642647–83644458	III	2	321	
TrWRKY135	chr2.jg6049	Tr2O:40646492–40650667	III	2	331	
TrWRKY136	chr5.jg4951	Tr5O:33015650–33019304	III	2	335	
TrWRKY137	chr13.jg1501	Tr5P:10308563–10310061	III	2	338	
TrWRKY138	chr7.jg7162	Tr7O:44915496–44917119	III	2	338	
TrWRKY139	chr16.jg1332	Tr8P:9621061–9623396	III	2	345	
TrWRKY140	chr12.jg6924	Tr4P:45351376–45352661	III	2	357	
TrWRKY141	chr12.jg6940	Tr4P:45450209–45451499	III	2	357	
TrWRKY142	chr6.jg2768	Tr6O:18717189–18720732	III	3	807	
TrWRKY143	chr5.jg5407	Tr5O:36181101–36185875	III	3	835	
TrWRKY144	chr5.jg5428	Tr5O:36242747–36247552	III	4	913	
TrWRKY145	chr5.jg5405	Tr5O:36173904–36179186	III	4	1,063	

Classification and phylogenetic analysis in white clover

An unrooted phylogenetic tree with 145 TrWRKY genes using neighbor joining methods (Fig. 1) was constructed to further explore the phylogenetic relationship of the WRKY transcription factor family in the white clover. This unrooted tree intuitively reflected the evolutionary status and grouping attribution of 145 members of the WRKY family. As shown in Fig. 1, the white clover WRKY proteins could be classified into three large groups (Group I-II) on the basis of the classifications of WRKYs in Arabidopsis. Specifically, the largest number of WRKY members in Group II was 84, while Group I and Group III had 37 and 24 members, respectively. In addition, Group II was further classified into five subgroups (SubGroup IIa-IIe). The most numerous subgroups were subgroup IIc, with 22 members. Next are IIb and IIe, with 21 and 20 members, respectively. Subgroup IId has 13 members. The subgroup with the lowest number of members in Group II is IIa, with only eight members. At the same time, the unrooted phylogenetic tree showed that the distribution of WRKY genes in white clover and Arabidopsis was highly consistent.

Figure 1 Phylogenetic analysis of white clover WRKY proteins.

The NJ tree was constructed from the amino acid sequences of TrWRKY using MEGA4 with 1000 bootstrap replicates. The white clover WRKY proteins were grouped into three groups (Group I, labeled with red solid circle, II, and III, labeled with blue black hollow circle), and the Group II was further divided into five subgroups (IIa labeled with purple solid triangle, IIb labeled with blue solid square, IIc labeled with pink solid diamond, IId labeled with green solid triangle, and IIe labeled with cyan hollow square).

Motif composition distribution analysis of TrWRKY proteins in white clover

To further our understanding of the molecular structure and function of the TrWRKY gene family in white clover, we have analyzed the conserved motifs of WRKY gene family members and found the same subgroup had similar motif composition. Ten individual motifs were identified by the MEME tool, revealing the distinct regions of TrWRKYs (Fig. 2, and Figs. S1–S4). As Fig. 2 shows, the TrWRKY genes in Group I have contained nine motifs in total, most of them (except TrWRKY001, 003, 005, 015, 035) contained motif 3 and motif 6, while motif 3 has “WRKY” residues (see Fig. S1), which confirmed WRKY domain in these TrWRKY genes. Motif 1 is present in TrWRKY genes without motif 3, which also contained “WRKY” residues (see Fig. S1). The results showed WRKY domain has diverged in white clover. In addition, there are double WRKY domains identified in Group I members, even three copies of WRKY domains, which is also supported by domain search results. Group II and III have shown similar results, each TrWRKY gene contained motif 1 or motif 3, even two motifs, which consisted of BLAST and domain search results (Figs. S2 and S3). Meanwhile, the results of conservation motif composition also supported the results of sequence similarity and phylogenetic tree analysis, demonstrating clear structural motif differences between the three group. For example, most members of Group I contain motif 3, while members of Group III contain motif 1 (see Fig. S4), similar appearance was also discovered in Group II, each subgroup has diff motif composition patterns, see Fig. S2. In each subgroup, the proteins harbor a similar number and type of motif, which suggested the functional similarities of these TrWRKYs.

Figure 2 Distribution of conserved motifs of TrWRKY genes Group I in white clover.

Cis-acting elements analysis of the TrWRKYs promoter

Promoter cis-elements influence the initiation of gene transcription. We performed a bioinformatics analysis to identify possible cis-elements in the promoter sequences of TrWRKYs. PlantCARE was used to identify putative cis-acting elements in the 1,000 bp upstream sequence of each TrWRKY gene promoter. A total of 15 stress response elements, consisting of TC-rich repeats (the cis-regulatory element for defense along with stress response), ACE (cis-regulatory element that engages in light response), LTR (cis-regulatory element that plays a role in low-temperature response), TCA-element (cis-regulatory element with a role in salicylic acid response), SARE (cis-regulatory element with a role in salicylic acid response), ABRE (cis-regulatory element associated with the abscisic acid response), AuxRR-core (cis-regulatory element with a role in auxin response), G-box (cis-regulatory element with a role in light response), CGTCA-motif (cis-regulatory element with a role in the MeJA-response), TGACG-motif (cis-regulatory element associated with MeJA-response), P-box (gibberellin-responsive element), GARE-motif (gibberellin-responsive element), WUN-motif (wound-responsive element), MBS (MYB binding site associated with drought-inducibility), and MRE (MYB binding site associated with light response), were identified (Fig. S5). All TrWRKYs had at least one stress response-linked cis-regulatory element. The cis-regulatory elements for hormone modulation consisting of CGTCA motifs, ABREs, AuxRR cores, P-boxes, TCA elements and TGA elements were also uncovered in numerous TrWRKY promoter regions. Overall, 64 TrWRKYs (44%) had more than one ABRE motif, which indicated the prospective abscisic acid response under stress conditions. Approximately 66 TrWRKYs (46%) had one or more CGTCA motifs that demonstrated the MeJA response potential, and the TCA element, TGACG motif, P-box, and AuxRR core were found in 30, 60, 12 and 8 TrWRKYs, respectively (Fig. S5). 68 G-box, 23 LTR, 29 MBS, and 25 TC-rich repeats were also found in TrWRKY promoter regions, which illustrated that these genes might play a role in cold, drought inducibility and defense responses.

Chromosome localization, gene duplication and Ka/Ks analysis of TrWRKY genes in white clover

To determine the evolution and expansion of WRKY genes, we used the MCScanX and Circos softwares to construct the distribution of WRKY genes across chromosomes. All TrWRKY genes were distributed across 16 chromosomes, but they were not uniformly located on these chromosomes, see Fig. 3. For example, the chromosomes TrChr1O, TrChr1P, TrChr8O and TrChr8P harbor more TrWRKY genes than other chromosomes, such as TrChr2O and TrChr2P. Based on BLAST results, there were 124 gene duplication events identified by MCScanX software, including 118 segment duplications (SD) and six tandem duplications (TD). The results have suggested chromosome doubling helps to bring about WRKY expansion in white clover, and distributions of TrWRKY were similar between some doubling chromosomes, for example, chromosomes TrChr6O and TrChr6P, chromosome TrChr5O and TrChr5P. However, there are some divergences between doubling chromosomes, such as chromosome TrChr3O and TrChr3P, chromosome TrChr7O and TrChr7P. The results suggested TrWRKY genes have undergone deep diversion in sub-genome evolution, some chromosomes have expanded TrWRKY members by gene duplication events, while some duplications have been purged, which caused some TrWRKY genes hots with numerous members clustering.

Figure 3 Chromosome distribution and expansion analysis of WRKY transcription factors in white clover.

Red lines show duplications between members of the Group I, purple lines show duplications between members of the subgroup IIa, green lines show duplications between members of the subgroup IIb, yellow lines show duplications between members of the subgroup IIc, greys lines show duplications between members of the subgroup IId, cyan lines show duplications between members of the subgroup IIe, and blue lines show duplications between members of Group III.

To estimate the divergence time of white clover WRKYs, synonymous (Ks) and nonsynonymous (Ka) substitutions between gene duplication pairs were calculated using the KaKs_Calculator in TBtool. When Ka/Ks <1, the genes experience purifying selection, which means the selection process could neutralize mutation to maintain the stability of the protein; in contrast, when Ka/Ks >1, the genes experience positive selection, which means great mutation happens in genes and eventually leads to a change in coded proteins. Our identified WRKY gene pairs had Ka/Ks values ranging from 0.06 to 0.89, proving that all of these genes experienced a purification selection process in white clover (Table S2).

Genetic regulation network analysis of white clover WRKY genes

Gene regulation networks (GRN) are increasingly used to explore the system-level functions of genes, we have reconstructed GRNs of TrWRKY and their interacting genes based public interaction database. The GRNs consisted of 349 genes and 463 interactions, as Fig. 4 shown. From GRNs, we have found most of TrWRKY have interacted with dozens of function genes, consisting of TrWRKY function on the transcription regulation process. For example, TrWRKY084 interacted with 37 genes, TrWRKY131 with 32 genes, and TrWRKY100 with 30 genes, the results indicated these TrWRKY genes played important roles in white clover lifespan. Gene Ontology (GO) annotation of these interacting genes was retrieved, and GO enrichment analysis was performed using the topGO package on the R platform. The results showed they were mainly distributed in the nucleus, see Fig. 5, which supported TrWRKY genes also functioned in the nucleus. In addition, molecular functions of these function genes were highly focused on transcription regulator activity, while they were mainly participating in the regulation of the transcription process, the results have confirmed the function of TrWRKY genes. It was notable that these genes were also enriched on terms “response to wounding” and “phosphorylay signal transduction system”, which are plant popular descriptions in response to abiotic stress, these results suggested TrWRKY genes probable function in response to abiotic stress.

Figure 4 Gene regulatory network analysis of TrWRKY genes and their interactions in white clover.

Gene regulatory network (GRN) of TrWRKY genes and their interactions were generated based Arabidopsis interactions, which was displayed with Cytoscape. Pink nodes correspond TrWRKY genes, while violet nodes correspond the genes interacted with TrWRKY genes, the cyan lines represented interactions in white clover.

Figure 5 Gene Ontology enrichment analysis of interaction genes with TrWRKY genes.

The GO enrichment analysis showed the involvement of interaction genes with TrWRKY genes in biological processes, molecular functions, and cellular components. Red dots represent GO terms from biological process (BP), green dots represent GO terms from molecular function (MF), while blue dots represent GO terms from cellular component (CC). Dot size represents the number of genes involving in the GO term, the X-axis is p-value of topGO enrichment analysis, with -log10 transformation, -log10 (p), while the Y-axis is GO terms.

Expression analysis of TrWRKY genes in response to the cold stress

In order to investigate TrWRKY genes function in response to abiotic stress, we have adopted our previous RNA-seq data under cold stress to assess their expression profiles. The white clover was treated with cold stress at 4 °C, and RNA-seq was analyzed at eight time points, including 0 h, 30 min, 1 h, 3 h, 6 h, 12 h, 24 h, and 72 h. All expressing TrWRKY genes (with FPKM value larger than 1) were collected, and their expressional value (FPKM) was grouped with a violin plot, and results showed TrWRKY genes were increased in response to cold stress, see Fig. S6. Especially, TrWRKY genes were rapidly activated at 30 min, and keeping a high expressional level in the following stages, the results suggested TrWRKY gene played critical roles in the early stage under cold stress. Expression profiles of these TrWRKY genes were clustered and displayed using the heatmap function, the results showed most of the TrWRKY genes were highly expressed at 30 min in response to cold stress, see Fig. 6. The finding consisted of violin plot analysis, which confirmed their rapid response to cold stress. Among these TrWRKY members, some were expressly high-expressing at 30 min, for example, TrWRKY017, 040, 049, 064, 065, 079, 085, 100, 102, 113, 138. Combining their hub-function in previous GRNs, such as TrWRKY100 and TrWRKY 113, they would be assigned rapid and critical regulation function in response to cold stress.

Figure 6 The expressional profiles of TrWRKY genes in response to cold stress.

The expressional profiles of TrWRKY genes were retrieved from RNA-seq database with accession numbers: PRJNA781064. There were eight time points, including 0 h, 30 min, 1, 3, 6, 12, 24, and 72 h, each time point has three biological replications. Mean expression levels (FPKM values) were measured by Salmon software (version 0.12.0), and they were displayed using ggplots package of R platform.

qRT-PCR validation of TrWRKY genes expression in response to cold stress

To validate the rapid response to cold stress, we have performed qRT-PCR analysis of seven TrWRKY genes with four time points, including 0 h, 30 min (0.5 h), 1 h, and 3 h. The qRT-RCR analysis results have confirmed their rapid and highly expressed in response to cold stress, see Fig. 7. All TrWRKY genes have shown a dramatic increase followed by a mild decrease in expression, the qRT-PCR and RNA-seq results are concordant, showing a similar pattern of TrWRKY genes expression in response to cold stress. These findings demonstrate that the TrWRKY genes are actively involved in the early response to cold stress and play critical roles in regulating gene expression in white clover.

Figure 7 qRT-PCR analysis of TrWRKY genes in response to cold stress.

The X-axis represent time points in response to cold stress, and Y-axis represent relative expression level of TrWRKY genes, which set expression level at “0 h” time point as 1. The expression level were calculated using the 2−ΔΔCT method as method section described.

Discussion

WRKY TFs are well indicated to regulate various physiological processes in plants, from plant growth, development, to respond to abiotic stress (Eulgem et al., 2000; Ülker & Somssich, 2004). With more and more plant genome sequencing accomplished, numerous WRKY TFs have been identified from many plants, with important roles in various physiological processes, especially their critical function in response to biotic or abiotic stress (Chen et al., 2015; Chen et al., 2016; Dong et al., 2019; Li et al., 2014; Rinerson et al., 2015; Singh et al., 2019). However, there is no report on the white clover, which is widely distributed on global earth.

In the present research, there are 145 WRKY TFs identified in white clover, whose assembly genome size is 841 Mb, the number is about two folds of Arabidopsis (72 members) (Abdullah-Zawawi et al., 2021). Based on previous research, the numbers of WRKY TFs are variations between different plants, which are not straightly related to genome sizes. For example, Arabidopsis contains 72 members in 119 Mb, cucumber contains 61 members in 225 Mb (Chen et al., 2020b), rice contains 128 members in 466 Mb (Abdullah-Zawawi et al., 2021), Medicago truncatula contains 93 members in 420 Mb (Song & Nan, 2014), and soybean contains 188 members in 994 Mb (Yu et al., 2016), etc. In these plants, rice and soybean have undergone whole genome duplication, which brought about WRKY TFs double in the genome. Therefore, white clover with the tetraploid genome, contained more WRKY TFs than plants with the diploid genome, such as Arabidopsis, cucumber. Besides, gene duplication is also considered with important role in the expansion of gene families, which are mainly classified as tandem duplication and segmental duplication based on duplication patterns. There were 118 segmental duplication events and six tandem duplication events in white clover, the results suggested segmental duplication played critical roles in the expansion of TrWRKY TFs. Meanwhile, the tandem duplication also has made minor contributions in the expansion of TrWRKY TFs compared to those with less WRKY members in the genome, such as cucumber (Chen et al., 2020b). Hence, gene duplication is the primary means of expansion and evolution of the white clover WRKY gene family.

Increasing reports have shown WRKY genes have played important roles in the regulation of signaling transduction, and transcription, especially in response to biotic or abiotic stress (Li et al., 2020; Waqas et al., 2019; Wei et al., 2021; Xie et al., 2018). In the present research, the genetic regulation network was reconstructed with 463 interactions. Most of the TrWRKY genes have more interacting genes, for example, TrWRKY065 interacts with 46 genes, TrWRKY086 with 42 genes, TrWRKY084 with 37 genes, TrWRKY131 with 32 genes, and TrWRKY018 with 31 genes, etc. According to GO enrichment analysis results, most genes had focused on the transcription process, regulation of the metabolic process, gene expression, etc. These results also confirmed that WRKY TFs functions were highly conservative in white clover. Meanwhile, molecular function results were also supported these findings, most interaction genes were annotated with DNA binding activity, implying that they are regulated by transcription factors. It is worth noting that the GO term “protein dimerization activity” was also enriched in GRN genes, which intensely suggested some TrWRKY genes interacted with each other to perform their function, for example, TrWRKY065 interacted with TrWRKY100, TrWRKY084 interacted with TrWRKY086, TrWRKY135 interacted with TrWRKY113, etc. Similar opinions were also reported in other plants, for example, the interaction of homoeologous WRKY18 and WRKY40 in Arabidopsis was confirmed with important roles in response to biotic stress (Abeysinghe, Lam & Ng, 2019); JrWRKY2 and JrWRKY7 were interacting and formatting into homodimers in response to abiotic stress and ABA treatment from Juglans (Yang et al., 2017); and WRKY genes were also found to form dimerization in rice (Cheng et al., 2019). These findings indicated function types of TrWRKY genes were consistent with other plants, which is needed more molecular experiments to validate. In addition, TrWRKY genes were characterized to participate in response to wounding, this result was also consisted with previous reports (Srivastava et al., 2022). In Arabidopsis, WRKY8 expression was wounding-induced and confirms involvement in basal defense (Chen, Zhang & Yu, 2010); Preferential expression of PsWRKY and its interaction with downstream genes in benzylisoquinoline alkaloids (BIAs) were possible involvement in response to wounding in Papaver somniferum (Mishra et al., 2013); OsWRKY53 was characterized in response to wounding stress, which was regulated by OsMKK4-OsMPK1 cascade, implying WRKY was involved in phosphorelay signal transduction pathway (Yoo et al., 2014). This conclusion was also testified in the present research, we have found that interactions of TrWRKY genes were focused on the phosphorelay signal transduction system, it has consisted of WRKY regulated by MAPK cascade in rice. The above analysis and discussion showed that the WRKY gene family could widely participate in various signaling pathways in response to biotic or abiotic stresses.

WRKY TFs have well documented in plant growth and various stress processes. Our previous RNA-seq has shown most of TrWRKY TFs were remarkably upregulated in response to cold stress (Zhang et al., 2022). All cold treatment samples were more highly expressed than the control sample, except the 12H sample, see Fig. S6. These results showed genes were down-regulated expressions at night, while TrWRKY genes were up-regulated by cold stress, and the final expressions were slightly upregulated. Other time points showed the expressions of most TrWRKY genes were sharply up-regulated under cold stress, especially at 30 min, there were 43 TrWRKY genes up-regulated at this time point, implying that these TrWRKY genes quickly response to cold stress. This finding was consistent with other plants, for example, in grapevine, Wang et al. (2014) have demonstrated 36 VvWRKYs were changed following cold exposure and identified 15 VvWRKYs in two or more cold expression datasets, intensely suggesting their key functions in response to cold stress; similar research has reported 10 WRKYs were strongly expressing during cold stress in tomato (Chen et al., 2015). These genes represented candidate genes for future functional analysis of WRKYs involved in the cold related signal pathways. By transgenic analysis, VvWRKY28 has greatly improved the tolerance of Arabidopsis to cold stress, bound to promote downstream genes, including (RAB18, COR15A, ERD10, PIF4, COR47, and ICS1), and promoted their expressions (Liu et al., 2022). PmWRKY57 was also identified to improve cold tolerance, and induce the expression levels of cold-response genes in Arabidopsis transgenic lines, including AtCOR6.6, AtCOR47, AtKIN1, and AtRCI2A (Wang et al., 2022). In addition, KoWRKY40 was demonstrated to involved in ICE-CBF-COR signaling pathway, functioned as an important regulator under cold stress (Fei et al., 2022). Meanwhile, CdWRKY2 from bermudagrass was also revealed as a positive regulator in cold stress by targeting CdCBF1 promoters and activating its expression, improving cold tolerance by opening the CBF-signaling pathway (Huang et al., 2022). In white clover, there were 19 target genes of TrWRKY genes identified with the AP2 domain, see Table S3, which was the conservative domain of CBF genes, implying TrWRKY genes probably conferred cold tolerance by regulated ICE-CBF-COR signaling pathway. However, the molecular mechanism of their regulation is still unclear, there are more experiments needed to explore their function in detail.

Conclusion

In summary, we have identified 145 WRKY genes in white clover, and the following analysis was performed: gene identification and classification, phylogenetic and motif composition distribution analysis, cis-acting elements analysis, chromosomal mapping and gene duplication analysis, gene expression analysis, Ka/Ks analysis, genetic regulation network analysis and Gene Ontology annotation, and quantitative real-time reverse transcription PCR (qRT-PCR) analysis. We found that the evolution and expansion of the WRKY gene family may be closely related to the replication of segment and tandem replication within the WRKY genes. Meanwhile, we identified WRKY genes that may be involved in response to cold stress. The results of RNA-seq demonstrated that WRKY gene expression was partially up-regulated in response to cold stress. qRT-PCR results directly revealed that the WRKY gene plays an important role in response to early cold stress in white clover. The results of this study suggest a foundation for further studies on the function of the WRKY gene family in response to biotic or abiotic stresses in white clover.

Supplemental Information

Supplemental Information 1 Primers used for qRT-PCR analysis of the TrWRKY genes.

Click here for additional data file.

Supplemental Information 2 Ka/Ks values of TrWRKY gene pairs in the white clover.

Click here for additional data file.

Supplemental Information 3 AP2 domains present in target genes of TrWRKY genes in white clover.

Click here for additional data file.

Supplemental Information 4 Raw data of qRT-PCR of TrWRKY genes.

Click here for additional data file.

Supplemental Information 5 Sequence logos for motifs identified from TrWRKY genes.

Click here for additional data file.

Supplemental Information 6 The distribution of conserved motifs from TrWRKY genes (Group Ⅱ) in white clover.

Click here for additional data file.

Supplemental Information 7 The distribution of conserved motifs from TrWRKY genes (Group III) in white clover.

Click here for additional data file.

Supplemental Information 8 Phylogenetic clustering and conserved protein motifs of TrWRKY genes in white clover.

Click here for additional data file.

Supplemental Information 9 The distribution of cis-acting elements in TrWRKY promoters.

Click here for additional data file.

Supplemental Information 10 The violin plot of all TrWRKY genes based their expression levels in response to cold stress.

Click here for additional data file.

We are grateful to high performance computing center of Harbin Normal University with supports on our analysis works.

Additional Information and Declarations

Competing Interests

Author Contributions

Data Availability

The authors declare that they have no competing interests.

Manman Li performed the experiments, analyzed the data, prepared figures and/or tables, authored or reviewed drafts of the article, and approved the final draft.

Xueqi Zhang performed the experiments, prepared figures and/or tables, and approved the final draft.

Tianxiang Zhang performed the experiments, prepared figures and/or tables, and approved the final draft.

Yan Bai analyzed the data, prepared figures and/or tables, and approved the final draft.

Chao Chen analyzed the data, prepared figures and/or tables, and approved the final draft.

Donglin Guo analyzed the data, prepared figures and/or tables, and approved the final draft.

Changhong Guo conceived and designed the experiments, authored or reviewed drafts of the article, and approved the final draft.

Yongjun Shu conceived and designed the experiments, performed the experiments, analyzed the data, prepared figures and/or tables, authored or reviewed drafts of the article, and approved the final draft.

The following information was supplied regarding data availability:

The raw data is available in the Supplemental Files.

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
