# Peer review of "Genome-wide analysis of the WRKY genes and their important roles during cold stress in white clover"

_PeerJ, doi:10.7717/peerj.15610_

## Round 0.1 · original submission · Major Revisions

Dear Dr. Su,
Thanks for submitting your work to PeerJ. Based on the comments from the reviewers, I hope to see the new version from you.
Thanks again
LIN

Reviewer 1 ·

Basic reporting

In this study, 145 WRKY genes have been identified in white clover, and the following analysis were performed: gene identification and classification, phylogenetic and motif composition distribution analysis, chromosomal mapping and gene duplication analysis, gene expression analysis, genetic regulation network analysis and Gene Ontology annotation, and quantitative real-time reverse transcription PCR (qRT-PCR) analysis. These findings are likely to be useful for further research on the functions of TrWRKY genes and their role in response to cold stress, which is important to understand molecular mechanism of cold tolerance in white clover, and improve its cold tolerance. But there are still some problems in the manuscript.

Experimental design

1. Line 49-51, “These WRKY TFs can bind to cis-acting element (named as W-box, (T)(T)TGAC(C/T)), and regulate expression of downstream target genes containing W-box in promotors.” Why not do cis-acting element analysis of TrWRKY genes in subsequent analysis? Please explain the reason or add cis-acting element analysis.
2. There were many gene duplication events that have been identified in TrWRKY genes. Moreover, WRKY genes play a variety of important roles in white clover. Why not do selective pressure analysis to see if positive selection occurs, to further explore its evolution. Please explain the reason or add cis-acting element analysis.

Validity of the findings

These findings are likely to be useful for further research on the functions of TrWRKY genes and their role in response to cold stress, which is important to understand molecular mechanism of cold tolerance in white clover, and improve its cold tolerance. But there are still some problems in the manuscript.

Additional comments

3. Line 54, “Arabidopsis” should be “Arabidopsis thaliana” the species appearing for the first time in the article should be given the Latin name. Please also checked throughout the manuscript.

4. Materials and Methods, please show the versions of softwares used in this study.

5. All gene names appearing in articles and diagrams should be italicized.

6. In the discussion section of the manuscript, a concluding statement should be added at the end of each discussion point.

Reviewer 2 ·

Basic reporting

1Line 121 (Alexa & Rahnenfuhrer 2019), the reference is in the wrong position in the text.
2There is an error in punctuation at the end of line 122.
3In Table 1, the general gene naming is based on the order in which family members appear on chromosomes. First, the genes on chromosome 1 are named according to their positions on chromosomes (such as TrWRKY001-TrWRKY025), and then the genes on chromosome 2 are named (such as TrWRKY026-TrWRKY045). Ask the author to consider whether to change it.
4Line 91 mentions "The WRKY Geneses Were Classified Into Groups Based on Similar WRKY Geneses in Arabia.", and WRKY phylogenetic trees of white clover and Arabidopsis should be provided. When classifying, we should not only refer to Arabidopsis sequences for classification, but also identify each sequence by grouping after mastering the sequence characteristics of classification. And in the later results and analysis, combined with motif analysis, the sequence characteristics of each group groups are expounded.

Experimental design

no comment

Validity of the findings

no comment

---

## Round 0.2 · accepted · Accept

Dear Dr. Shu,

Thanks for considering PeerJ. Congratulations, there are two reviewers who recommend accepting your work.

Warm regards
LIN

Reviewer 1 ·

Basic reporting

The manuscript has been improved greatly after the revision, and can be accepted now.

Experimental design

no comment

Validity of the findings

no comment

Additional comments

no comment

Reviewer 2 ·

Basic reporting

The revised manuscript is acceptable

Experimental design

The revised manuscript is acceptable

Validity of the findings

The revised manuscript is acceptable

Additional comments

The revised manuscript is acceptable